# Dosimetric Comparison of Exposure Pathways to Human Organs and Tissues in Radon Therapy

**DOI:** 10.3390/ijerph182010870

**Published:** 2021-10-15

**Authors:** Werner Hofmann, Herbert Lettner, Alexander Hubmer

**Affiliations:** 1Biological Physics, Department of Chemistry and Physics of Materials, University of Salzburg, Hellbrunner Str. 34, 5020 Salzburg, Austria; herbert.lettner@plus.ac.at (H.L.); alexander.hubmer@plus.ac.at (A.H.); 2Radiological Measurement Laboratory, Department of Chemistry and Physics of Materials, University of Salzburg, Hellbrunner Str. 34, 5020 Salzburg, Austria

**Keywords:** radon therapy, thermal water, radon inhalation, biokinetic models, organ dose calculations

## Abstract

Three therapeutic applications are presently prescribed in the radon spas in Gastein, Austria: exposure to radon in a thermal bath, exposure to radon vapor in an exposure chamber (vapor bath), and exposure to radon in the thermal gallery, a former mine. The radiological exposure pathways to human organs and tissues in these therapeutic radon applications are inhalation of radon and radon progeny via the lungs, radon transfer from water or air through the skin, and radon-progeny deposition on the skin in water or air. The objectives of the present study were to calculate radon and radon-progeny doses for selected organs and tissues for the different exposure pathways and therapeutic applications. Doses incurred in red bone marrow, liver, kidneys, and Langerhans cells in the skin may be correlated with potential therapeutic benefits, while doses to the lungs and the basal cells of the skin indicate potential carcinogenic effects. The highest organ doses among the three therapeutic applications were produced in the thermal gallery by radon progeny via inhalation, with lung doses of 5.0 mSv, and attachment to the skin, with skin doses of 4.4 mSv, while the radon contribution was less significant. For comparison, the primary exposure pathways in the thermal bath are the radon uptake through the skin, with lung doses of 334 μSv, and the radon-progeny attachment to the skin, with skin doses of 216 μSv, while the inhalation route can safely be neglected.

## 1. Introduction

In the traditional radon spas Bad Gastein and Bad Hofgastein, Austria, natural radon-rich water in springs and high radon levels in ambient air in the thermal gallery have been successfully applied in the past for the treatment of various rheumatic diseases, most notably, ankylosing spondylitis [1,2]. Although the physiological and cellular mechanisms induced by the very low radiation doses resulting from the radon and radon-progeny exposures are still not yet fully understood, several studies in the Gastein treatment facilities have proven the effectiveness of therapeutic radon treatment [3,4,5,6,7]. A comprehensive list of recommended indications for radon treatment and a summary of clinical trials with radon applications have recently been published by Maier et al. [8].

In contrast to its reported beneficial role in radon therapy, inhalation of environmental radon and its progeny is commonly associated with the potential induction of bronchial carcinomas, even at low radon levels as in homes [9]. Thus, the therapeutic application of radon is inherently associated with potential harmful effects.

Three primary therapeutic regimes are presently applied in Bad Gastein and Bad Hofgastein: (i) exposure to radon in a thermal bath, (ii) exposure to radon vapor in a specially designed exposure chamber (“vapor bath”), and (iii) exposure to radon in the thermal gallery (“Gasteiner Heilstollen”), a former gold and silver mine. In the thermal bath, patients sit in a bathtub (37–40 °C) for a period of 20 min (on average ten treatments) with a radon activity concentration in water of about 900 kBq m^−3^. In the vapor bath, patients are exposed to radon vapor (32–41.5 °C and 70–99% RH) in a small exposure chamber for a period of 20 min (on average ten treatments) with radon activity concentrations ranging from 30 to 200 kBq m^−3^ (average value: 90 kBq m^−3^). Exposure conditions in the vapor bath are similar to those in the thermal gallery, except that the head of the patient remains outside of the exposure chamber, thereby significantly reducing the inhalation of radon and its progeny. In the thermal gallery, patients are exposed to radon and radon progeny in mine air for 60 min (on average ten treatments) at an average radon activity concentration of about 45 kBq m^−3^ (maximum 160 kBq m^−3^) at 37–41.5 °C and 70–99% RH.

The radiological exposure pathways to human organs and tissues for the three therapeutic treatment schemes are inhalation of radon and radon progeny via the lungs, radon transfer from water or air through the skin, and radon-progeny attachment to the skin in water or air. Incorporated radon and radon progeny are subsequently distributed among the organs and tissues of the human body via the blood stream.

Several physiologically based pharmacokinetic (PBPK) models for the distribution of inhaled radon throughout the human body have been developed in the past, e.g., by Peterman and Perkins [10], Khursheed [11], Sakoda et al. [12,13], Leggett et al. [14], and Hofmann et al. [15]. In addition, Khursheed [11] and Sakoda et al. [12] provided information on dose conversion coefficients per unit radon activity concentration for a wide range of organs and tissues.

Since lung cancer risk is the dominant detrimental biological effect of inhaled radon progeny, the primary target for dose calculations are the lungs, especially the bronchial region. Among the different radon-progeny lung-dosimetry models developed in the past few decades [16], bronchial dose conversion coefficients per unit Working Level Month (WLM) were recently reported by Füri et al. [17], Harley [18], Hofmann and Winkler-Heil [19,20], Winkler-Heil et al. [21], and Marsh et al. [22]. Considering the subsequent distribution of inhaled radon progeny throughout the human body, dose conversion coefficients for several organs and tissues have been published by Al-Jundi et al. [23], Brudecki et al. [24], Kendall and Smith [25,26], and Marsh et al. [27].

The distribution of radon in different organs arising from the transfer of radon from thermal water through the skin was modeled by Sakoda et al. [13] and Hofmann et al. [15] by extending the PBPK model of Leggett et al. [14] with additional skin compartments.

An additional contribution to the skin dose is produced by the attachment of radon progeny to the skin of patients in the thermal bath and the vapor bath [28,29,30]. In contrast to the other exposure pathways, radon-progeny surface activities affect exclusively the skin but not the other organs. Besides the stimulation of an immune response in the Langerhans cells (LC) of the skin, alpha particles emitted from radon progeny deposited on the skin surface may eventually cause skin cancer in the basal cells (BC) of the skin [31].

The therapeutic response following radon exposures has traditionally been attributed to the radiation doses received by specific organs and tissues of the human body. Consequently, any methodological interpretation of therapeutic mechanisms triggered by the radon exposures requires a thorough analysis of the radiation doses incurred by sensitive organs or tissues in the different exposure pathways applied in the Gastein radon therapy facilities.

While radon and radon-progeny activity concentrations in water and inhaled air are already available for practically all exposure conditions, measurements of radon uptake through the skin in vapor bath and thermal gallery are still missing. Likewise, except for the attachment of radon progeny to the skin of patients in the thermal and vapor baths [29,30], only a few dose calculations for different organs and tissues via radon and radon-progeny inhalation, as well as radon uptake through the skin, have been published in the past for the specific Gastein exposure conditions [29,30,32,33,34]. Thus, the objectives of the present study were (i) to complete the experimental radiological database by measuring the radon transfer through the skin in the vapor bath, (ii) to calculate radon and radon-progeny doses for selected organs and tissues of the human body for the radiological conditions characterizing the different therapeutic exposure pathways, and (iii) to compare selected organ doses incurred in the different treatment schemes as a basis for the application of specific therapeutic regimes.

## 2. Materials and Methods

### 2.1. Experimental Study of Radon Transfer through the Skin in the Vapor Bath

The experimental methods of the exhalation measurements in the vapor bath were the same as for the thermal bath, and have already been described in detail by Lettner et al. [35] and Hofmann et al. [15]. Thus, only a few salient features of the experimental methods will be discussed here for the understanding of the measured exhalation curves.

In the vapor bath, volunteers were exposed to hot radon vapor in a small chamber (volume approximately 1 m^3^) with an average radon activity concentration of about 90 kBq m^−3^ at 32–41.5 °C and 70–99% RH, with the head remaining outside of the exposure chamber. Volunteers inhaled air through a tightly fitting breathing mask with an inlet and an outlet tube, thus permitting inhalation and exhalation through the mouth. While the volunteers inhaled practically radon-free outdoor air (average radon activity concentration of about 10 Bq m^−3^), they exhaled radon either into room air or in pre-specified time intervals (about 4 min) into a radon-tight aluminum foil-synthetic bag for subsequent analysis. After a single exhaled breath, which is sufficient to obtain measurable radon activity concentrations, the radon-containing plastic bag was disconnected from the breathing mask and replaced by a new one. This sampling protocol allowed the measurement of exhaled radon activity concentrations in approximately 4 min intervals during the 20 min exposure period. These exhalation measurements were then continued for another 20 min after the volunteer left the exposure chamber to track the decrease in the radon activity concentration in the exhaled air. For the measurement of radon activity concentrations in the exposure chamber, radon vapor was transferred to the plastic containers via a pump, with filters before and after the pump. The radon-containing plastic containers were then analyzed in the laboratory in Salzburg by alpha liquid scintillation counting with Lucas cells of 250 mL and the Pylon AB-5 radon monitor (Pylon Instruments).

Two exhalation measurements were performed in the vapor bath with a male volunteer (age: 30 years, height: 187 cm, mass: 83 kg, body surface area: 2.08 m^2^, lung volume: 3.54 l, respiratory minute volume: 466 l h^−1^) who already participated in the previous study for the thermal bath (subject 6) [15]. Radon activity concentrations in the exposure chamber during both experiments ranged from 171.9 to 306.0 kBq m^−3^, i.e., significantly higher than the commonly assumed typical average value of 90 kBq m^−3^ observed during routine therapeutic exposures. While the exhalation curves in the thermal bath reflected the diffusion of radon from water through the skin, the corresponding exhalation curves in the vapor bath referred to the radon transfer from humid air through the skin covered by a thin water layer due to sweat and vapor deposition.

The two exhalation curves measured in this study, normalized to the corresponding mean radon activity concentrations, during the 20 min exposure and 20 min resting phases, are plotted in Figure 1 and compared with the corresponding exhalation curve for the thermal bath [15].

The comparison of the exhalation measurements demonstrates that the shapes of the exhalation curves in the vapor bath were similar to those for the thermal bath, except that the maximum values at the end of the 20 min exposures were lower, if normalized to the same radon activity concentration in air or water. The similarity of the shapes of the exhalation curves indicates that the operating mechanisms for the radon transport through the skin were the same for both water and air, although not their relative magnitudes. This further suggests that transfer rates can be scaled by the measured maximum radon activity concentrations after 20 min, which yielded an average ratio of the transfer rates in the vapor bath to those in thermal water of about 0.32. The higher permeability of the skin in hot water relative to hot vapor may be attributed to the opening of the pores, thereby facilitating the transport of radon through the skin. Since the radon activity concentration in the exposure chamber of the vapor bath was about one order of magnitude smaller than that in thermal water, i.e., 90 kBq m^−3^ vs. 900 kBq m^−3^, the radon transfer in the vapor bath during the treatment was lower by a factor of about 0.03.

Compared to the relatively smooth exhalation curve for the thermal bath, where the statistical error of the individual measurements was estimated to range for ± 5 to 15% [15], the exhalation curves for the vapor bath exhibited significant fluctuations, primarily during the exposure phase. Two factors may have contributed to these increased fluctuations relative to the thermal bath. Wearing a tight face mask and exhaling hot and humid air while sitting in a narrow, hot, and humid exposure chamber for 20 min can exert considerable physical stress, which may have affected the breathing pattern of the volunteer. Furthermore, the radon activity concentration in the room air of the vapor bath (3.5 kBq m^−3^) was two to three orders of magnitude higher than that of the inhaled outdoor air of 10 Bq m^−3^. Thus, even a minimal change in the air-tightness of the mask during inhalation could have increased the radon activity concentration in the subsequent exhaled breath. For comparison, this effect could safely be neglected for the thermal bath because of the much lower radon activity concentration in the ambient air.

Unfortunately, corresponding exhalation measurements in the thermal gallery could not be conducted for various reasons. In the thermal gallery, the inhalation of cool outside air with low radon activity concentrations during the exposure was technically not possible. Without the inhalation of outside air, however, breathing of such hot and humid air through a face mask over a period of 1 h would hardly have been bearable for a volunteer. Furthermore, concomitant inhalation of high ambient radon activity concentrations would have interfered with radon transfer through the skin, as the radon activity concentrations in the exhaled air would be primarily determined by the inhaled radon. Thus, in the absence of pertinent experimental data, it was assumed that the transfer coefficients for radon through the skin in the thermal gallery were the same as in the vapor bath, i.e., radon transfer rates were scaled by the corresponding radon activity concentrations.

### 2.2. Organ Dosimetry Model for Radon Inhalation

Dose conversion coefficients for different organs and tissues of the human body for radon inhalation have been published by Khursheed [11] and Sakoda et al. [12]. These dose conversion coefficients are based on the continuous inhalation of radon and thus refer to steady-state conditions of the radon activity concentrations in the different organs. Previous calculations demonstrated that radon activity concentrations in most organs reached steady-state conditions already within a few minutes, except for the fat-containing organs, e.g., red bone marrow, which exhibited a slower increase [12,36]. Since patients inhaled enhanced radon levels in all therapy facilities prior to the therapeutic exposures, specifically in the thermal gallery, steady-state activity concentrations may have already existed in most organs at the start of the actual exposures. Furthermore, radon activity concentrations dropped quickly in most organs. Thus, organ-specific steady-state dose conversion coefficients were assumed to reflect the realistic exposure conditions in the present study. Only in the special case of the fatty tissues, where radon activity concentrations rise much slower and may even not reach steady-state conditions during the exposure time [12,36], published dose conversion coefficients may overestimate resulting doses. However, these activity deficits during the exposure times of 20 min and 60 min, respectively, may partly be compensated by the observation that radon activity concentrations still persist in these tissues for another couple of hours [15,36]. Thus, in the absence of pertinent information on the time course of radon activity concentrations in the fat-containing organs, the dose conversion coefficients of Khursheed [11], assuming steady-state conditions, were considered as realistic approximations of the actual exposure situations. Since the biokinetic model of Khursheed [11] does not include a skin compartment, corresponding results of steady-state radon activity concentrations in the skin provided by Sakoda et al. [12] were used for the dose calculations.

### 2.3. Organ Dosimetry Model for Radon Uptake through the Skin

The uptake of radon from thermal water through the skin, its transfer to blood, its subsequent distribution among human organs and tissues via the blood stream, and its final exhalation, have already been described by the transfer and organ distribution model of Hofmann et al. [15]. In this model, based on the biokinetic model of Leggett et al. [14] for the inhalation of radon, human organs and tissues are represented by eight compartments, of which three, i.e., fat, bone, and skin, are each further divided into two sub-compartments. For the simulation of the transfer of radon from the thermal water through the skin into the venous blood pool, an additional skin compartment was added by Sakoda et al. [13] and Hofmann et al. [15]. Since the skin consists of the epidermis, the dermis with very little blood supply and low in fat, and the adipose hypodermis or subcutaneous tissue, which is rich in blood vessels and fat, the skin compartment was further subdivided into a dermal skin (DS) compartment and a subcutaneous skin (SS) compartment [15]. Because of its much higher blood flow and fat content, the radon transfer from water to blood in the skin takes place primarily in the adipose subcutaneous tissue layer. Thus, the pathway of radon from thermal water to venous blood is water-dermal skin-subcutaneous skin-venous blood. For the simulation of the radon transfer from ambient air through the skin in the vapor bath and thermal gallery, all model parameters were assumed to be the same as for the thermal bath, except for the transfer rates from air to dermal skin. This difference between diffusion of radon from air through a wet skin and diffusion of radon from water through the skin was experimentally determined by comparing the corresponding radon exhalation curves (see Section 2.1).

The time-dependent radon activity concentrations in different organs and tissues, determined by the transfer and organ distribution model, served as the basis for subsequent organ dose calculations. Integration of the organ-specific radon activity concentration curves from start to end of exposure (20 min for thermal and vapor baths and 1 h for the thermal gallery) and during the post-exposure period until all radon nuclides have decayed, provides the total radon activity accumulated in a given organ. This total radon activity together with the volume and mass of that organ [14] allow then the calculation of the total dose delivered by emitted radon alpha particles during a treatment session.

### 2.4. Organ Dosimetry Model for Radon-Progeny Inhalation

Previous dose calculations for the vapor bath and the thermal gallery were based on the assumption that radon-progeny inhalation is the primary cause of the observed therapeutic response [32,33,37]. Current dose estimates for individual organs and tissues for the continuous inhalation of radon progeny in the vapor bath and thermal gallery [36] are based on the dose calculations of Hofmann [33], derived from the initial calculations of Pohl [38]. Corresponding organ doses for radon-progeny exposure in the thermal bath have been published by Hofmann et al. [34]. Since radon and radon-progeny activity concentrations in the vapor bath and the thermal gallery have changed in recent years, organ and tissue doses presented in this study were based on currently measured average activity concentrations.

Dose conversion coefficients for different organs and tissues of the human body, published by Brudecki et al. [24] and Kendall and Smith [25], refer to radon-progeny inhalation over a longer period of one year, primarily for radiation protection purposes. Thus, these dose conversion coefficients refer again to the continuous inhalation of radon progeny and thus are based on steady-state conditions of the radon-progeny activity concentrations in the different organs and tissues.

Unfortunately, no information is currently available on the time course of the radon-progeny activities in the different organs for short exposure times, such as 20 to 60 min in radon therapy. Since the transport of radon progeny deposited on lung airway surfaces to other organs and tissues via the lungs or the gastro-intestinal (GI) tract takes much longer than that for the noble gas radon, the steady-state approximation may lead to an overestimation of the doses incurred in different organs for short exposure periods. On the other hand, patients in the thermal gallery inhaled enhanced radon-progeny levels on their way to the therapy station, so that steady-state activity concentration may have already existed in most organs at the start of the actual exposures. Furthermore, organs were further irradiated by alpha particles after the end of the exposures due to the continuing decay of the alpha activities, which may again have persisted for several hours. Thus, in the absence of more detailed information on the temporal behavior of the transfer of radon progeny among the organs of the human body during and after inhalation, the dose conversion coefficients provided by Kendall and Smith [25] were applied in the dose calculations. Since the biokinetic model of Kendall and Smith [25] does not contain a skin compartment, dose calculations for the skin were based on the inhalation dose coefficient of Brudecki et al. [24].

### 2.5. Lung Dosimetry Model for Radon-Progeny Inhalation

The stochastic dosimetry code IDEAL-DOSE [19,39] comprises four stochastic models: (1) The asymmetric stochastic airway generation model represents the inherent asymmetry and variability of the human airway system in terms of distributions of airway diameters, lengths and branching, and gravity angles [40]. (2) The stochastic particle deposition model IDEAL simulates the random walk of inhaled radon progeny through the random, asymmetric airway structure, applying Monte Carlo techniques [41]. (3) In the stochastic bronchial clearance model, mucus clearance velocities in individual airways of the asymmetrically branching airways are related to their respective diameters [42]. (4) Finally, in the stochastic cellular dosimetry model, the depths of basal- and secretory-cell nuclei in bronchial airways are correlated with their diameters, and basal- and secretory-cell doses are weighted by their relative frequencies [43]. As a result of methodological and computational differences, dose-exposure conversion coefficients for the IDEAL-DOSE model are consistently lower than those based on the ICRP Human Respiratory Tract Model (HRTM) [22,44] by about 20–30% [21].

### 2.6. Skin Dosimetry for Radon-Progeny Attachment to the Skin

The measurement of radon-progeny activities attached to skin surfaces, and the resulting dose calculations for the Langerhans cells, represented by the epidermal layer, in thermal and vapor baths have already been reported in Tempfer et al. [29,30]. Radon-progeny activities on selected skin surfaces were experimentally determined by alpha spectrometry as a function of time in 1 m intervals during the exposure and post-exposure periods. The corresponding distribution of radon progeny throughout the thickness of the skin was obtained by successively removing 5 μm-thick cell layers, revealing a roughly exponential activity distribution in the upper layers of the skin. By integrating the dose distribution as a function of depth over the whole thickness of the epidermis, average skin doses could be determined for both thermal and vapor baths. Because of the similarities of the radon-progeny exposures in vapor bath and thermal gallery [45], skin dose calculations for the thermal gallery were based on the surface activity measurements in the vapor bath, considering differences in radon activity concentrations and exposure times. In the case of the radon-progeny exposure of the skin in the thermal gallery, however, the exposed skin surface area may have been somewhat smaller as patients were lying on their back and were partly covered by swimsuits.

Recently, Sakoda et al. [28] developed a model for the skin deposition velocity of radon progeny in thermal water and air, based on the experimental deposition data of Tempfer et al. [30], which allows the calculation of radon-progeny activities on skin surfaces for different exposure conditions. The result of their analysis was that the deposition velocities of radon progeny were 0.08 m h^−1^ (attached) and 8 m h^−1^ (unattached) in air, compared to 0.024 m h^−1^ (both attached and unattached) in water.

## 3. Results of Dose Calculations

To facilitate comparison between the different exposure pathways and between different treatment facilities, all calculated doses were based on an average number of 10 treatments for each patient, with 20 min per session for thermal bath and vapor bath, and 60 min for the thermal gallery. Likewise, the resulting doses refer to typical average radon and radon-progeny exposure conditions for each exposure pathway, derived from a wide range of individual measurements. These average radiological input parameters used for the present dose calculations are compiled in Table 1. Thus, if more or fewer treatments were prescribed for individual patients or actual radon and radon-progeny activity concentrations at a given time of exposure deviated from these average values, corresponding patient-specific doses could be obtained by multiplying the calculated doses by the ratio of the number of treatments or by the ratio of the radon and radon-progeny activity concentrations.

Dose calculations presented in subsequent Table 2, Table 3, Table 4, Table 5, Table 6, Table 7 and Table 8 refer to adult patients, based on average values for female and male biokinetic parameters and organ masses. For radon and radon-progeny inhalation, respiratory minute volumes for sitting awake were assumed in thermal and vapor baths and for resting in the thermal gallery [44].

### 3.1. Radon Inhalation

For the calculation of organ doses arising from the inhalation of radon in ambient air, the following activity concentrations were assumed:

#### 3.1.1. Thermal Bath

The selected radon activity concentration of 250 Bq m^−3^ represented an average value of several measurements in different bathrooms and at different times by Lettner et al. [46] and Hofmann et al. [34], ranging from 190 to 280 Bq m^−3^.

#### 3.1.2. Vapor Bath (Room Air)

A typical average value during the exposure of the patients of 3.5 kBq m^−3^ was derived from several unpublished measurements, ranging from 2.80 to 3.74 kBq m^−3^, which were lower than the value of 5 kBq m^−3^ reported in an earlier study [30]. Note that radon activity concentrations in room air are on average only 60 Bq m^−3^ when the radon supply to the exposure chamber is turned off.

#### 3.1.3. Thermal Gallery

Typical average radon activity concentrations published in the past were 44 kBq m^−3^ [47,48], 45 kBq m^−3^ [30], and 50 kBq m^−3^ [49], while our own regular measurements over a twenty-year period yielded an average value of 59 kBq m^−3^. Based on these measurements, an average value of 50 kBq m^−3^ was chosen for the dose calculations. Note that the actual radon levels can vary considerably for a specific exposure as a result of varying ventilation conditions and barometric pressure [50,51].

Equivalent doses to selected organs resulting from the inhalation of radon in the thermal bath (250 Bq m^−3^), vapor bath (3.5 kBq m^−3^), and thermal gallery (50 kBq m^−3^), based on the dose conversion coefficients of Khursheed [10], are listed in Table 2. These dose conversion coefficients refer to annual equivalent doses for an indoor radon activity concentration of 200 Bq m^−3^ with 22 h d^−1^ indoor occupancy. Since no breathing rates were reported by Khursheed [11], a breathing rate of 0.9 m^3^ h^−1^ was assumed for the population exposure, as suggested by Kendall and Smith [25]. Differences in radon activity concentrations, exposure times, and respiratory minute volumes in the thermal bath, vapor bath, and thermal gallery relative to the published dose conversion coefficients of Khursheed [11] were considered by multiplicative scaling factors of the respective parameter ratios. The same procedure was applied to the dose conversion coefficients of Sakoda et al. [12] for the skin.

The organs receiving the highest doses were the lungs as the port of entry into the human body, and the red bone marrow because of the relatively high solubility of radon in fatty tissue. Note that the dose to the lungs was based on the total number of disintegrations in lung air and blood, thus assuming that the decay energy as well as the sensitive target cells were distributed uniformly throughout the lungs. The dose to the kidneys was also representative of stomach, small intestine, colon, gonads, brain, bladder, and muscle.

For comparison, Sakoda et al. [12] reported the following ratios of radon activity concentrations after continuous inhalation of 1 Bq m^−3^ in selected organs between the model of Khursheed [11] and their study: 20 (lungs), 1.53 (liver), 0.86 (kidneys), and 0.83 (RBM). This indicates that, with the exception of the lungs, both models yield comparable organ doses. In the case of the skin, dose calculations were based on Sakoda et al. [12], as the biokinetic model of Khursheed [11] does not contain a skin compartment. In the absence of any pertinent information about the radon distribution within the skin compartment with relation to the distribution of Langerhans and basal cells, a uniform dose distribution was assumed, leading to the same doses to both cell types.

### 3.2. Radon Transfer through Skin

The calculation of organ doses as a result of the transfer of radon from water or air via the skin into the human body was based on the following activity concentrations:

#### 3.2.1. Thermal Bath

Based on published average values of 950 kBq m^−3^ [30] and recent measurements of 865 kBq m^−3^, ranging from 720 to 960 kBq m^−3^ [15], an average radon activity concentration of 900 kBq m^−3^ was assumed for the dose calculations.

#### 3.2.2. Vapor Bath (Exposure Chamber)

Reported average radon activity concentrations ranged from 60 kBq m^−3^ (40–140 kBq m^−3^) [49] to 90 kBq m^−3^ (30–200 kBq m^−3^) [30], and 140 kBq m^−3^ [37]. Considering the significant temporal fluctuations, an average value 90 kBq m^−3^ was adopted as a justifiable compromise.

#### 3.2.3. Thermal Gallery

An average radon activity concentration of 50 kBq m^−3^ was assumed as discussed above (see Section 3.1.3).

While the radon levels in thermal water are relatively constant, ambient radon activity concentrations in vapor bath and thermal gallery can vary significantly due to temporal fluctuations in radon production in the underground wells and meteorological variations [50,51].

In the thermal bath as well as in the vapor bath, the heads of the patients remain outside of the water or of the exposure chamber, thus reducing the total surface area of the human body by about 7.5% [52]. Furthermore, the bodies of the patients in all therapeutic applications are partly covered by bathing suits or are lying on their back in the thermal gallery, thereby further decreasing the effective surface area of the body for the transfer of radon through the skin.

Equivalent doses to selected organs resulting from the uptake of radon from water in the thermal bath (950 Bq m^−3^), vapor bath (90 kBq m^−3^), and thermal gallery (50 kBq m^−3^), based on previous exhalation measurements in the thermal bath [15] and the new measurements in the vapor bath presented in this study (see Section 2.1) are listed in Table 3, based on average radon activity concentrations for male and female volunteers. Corresponding doses for the thermal gallery are extrapolated from the vapor bath data, scaled by the ratio of the radon activity concentrations and exposure times.

Doses to Langerhans and basal cells were based on the radon activity concentration in the dermal skin compartment of the biokinetic model of Hofmann et al. [15], where the skin is subdivided into two layers, a dermal skin and a subcutaneous fat compartment. The assumption of uniform dose and target cell distributions in the dermal skin layer leads again to the same doses to Langerhans and basal cells. For comparison, the skin in the model of Sakoda et al. [13] is represented by only one compartment, which conceptually resembles the dermal skin compartment. Consequently, both models predict similar skin doses.

### 3.3. Radon-Progeny Inhalation

The calculation of organ doses arising from the inhalation of short-lived radon progeny requires information on radon activity concentration, equilibrium factor (F), or radon-progeny activity concentration expressed as Working Level (WL), activity median diameters (AMD) of attached and unattached progeny, and unattached fraction (f_p_) of the potential alpha energy concentration (PAEC).

#### 3.3.1. Thermal Bath

An average radon activity concentration of 250 Bq m^−3^ was assumed as discussed above (see Section 3.1.3).

Based on our own unpublished measurements and published data [34,46], an equilibrium factor F = 0.20 was adopted as a typical average value.

Activity median diameters (AMD) of 290 nm for attached radon progeny and of 1.1 nm for the unattached fraction were derived from own measurements [33].

Measurements of the unattached fraction (f_p_) of the PAEC with a diffusion battery indicated an average value of 25% [34].

#### 3.3.2. Vapor Bath (Room Air)

Radon progeny in room air is produced by radon and radon progeny in the exposure chamber, as the opening in the exposure chamber for the head is not tight enough to prevent radon and radon progeny from escaping the chamber. Thus, radiological parameters of radon progeny in room air are comparable to the corresponding parameters in the exposure chamber.

An average value of the radon activity concentration during the exposure of the patients of 3.5 kBq m^−3^ was adopted as discussed above (see Section 3.1.2).

For the equilibrium factor, Just [49] reported a value of 0.1, which is consistent with our own measurements of F = 0.11 in the exposure chamber.

Since radon-progeny exposure conditions in the vapor bath are practically the same as in the thermal gallery [45,48], the same AMD of 230 nm was assumed for the inhalation calculations. For the AMD of the unattached fraction, a typical value of 1 nm was assumed [38].

In the absence of direct measurements, an f_p_-value of 8% was considered as a best estimate, based on a typical value of 10% in rooms [53], slightly reduced by the presence of additional water-vapor aerosols.

#### 3.3.3. Thermal Gallery

For the radon activity concentration in the thermal gallery, an average value of 50 kBq m^−3^ was chosen for the dose calculations (see Section 3.1.3).

Typical equilibrium factors published in the past were 0.4 [48], 0.48 [49], 0.5 [45], 0.53 [37], 0.54 [51], and 0.69 [47], while our own regular measurements over a twenty-year period yielded an average value of 0.47, ranging from 0.35 to 0.56. Based on these measurements, an average value of 0.5 was chosen for the dose calculations.

In addition to radon activity concentrations and F-values, radon-progeny activity concentrations in Working Level (WL) were also measured in the thermal gallery. Based on average values of 6.9 WL (7 measurements), ranging from 3.13 to 17.4 [51] and 6.7 WL (14 of our own measurements over a twenty-year period), an average value of 6.8 WL was adopted for the dose calculations. Note that an average radon activity concentration of 50 kBq m^−3^ and an average F-value of 0.5, as derived above for the dose calculations, yield the same WL-value.

The high relative humidity in the thermal gallery did not permit direct measurements of the attached size distribution of the short-lived radon progeny in the thermal gallery. The only currently available information on radon-progeny size distribution was provided by Wallner et al. [54], who measured the activity size distribution of ^212^Pb and ^210^Pb with a cascade impactor and subsequent liquid scintillation counting. Since Butterweck et al. [55] observed that AMD values for ^212^Pb were similar to those of the radon progeny ^214^Pb/^214^Bi, the AMD value of 230 nm for ^212^Pb reported by Wallner et al. [54] was also adopted for the radon progeny. This value is practically identical with an AMD of 228 nm measured by Butterweck et al. [55] in the Postojna tourist cave for ^214^Pb/^214^Bi activities. For the AMD of the unattached fraction, a typical value of 1 nm was assumed.

The only measurement of f_p_ in the thermal gallery was reported by Landstetter [51], who determined the unattached fraction by the continuous measurement of the aerosol particle concentration Z during a period of 2 days, utilizing the inverse relation between f_p_ and Z [56]. The relatively low value of 3.6%, was attributed to the high particle concentration in the thermal gallery. Thus, a value of 4% was adopted for the dose calculations.

Organ doses resulting from the inhalation of radon progeny in the thermal bath, vapor bath, and thermal gallery are listed in Table 4, based on the dose conversion coefficients of Kendall and Smith [25]. These dose conversion coefficients refer to annual equivalent doses for an indoor radon activity concentration of 200 Bq m^−3^ with F = 0.4, a breathing rate of 0.9 m^3^ h^−1^, and 22 h d^−1^ indoor occupancy.

Differences in radon activity concentrations, equilibrium factors, exposure times, and respiratory minute volumes in the thermal bath, vapor bath, and thermal gallery relative to the published dose conversion coefficients of Kendall and Smith [25] were considered by multiplicative scaling factors of the respective parameter ratios. In the case of the skin, the same procedure was applied to the dose conversion coefficients of Brudecki et al. [24]. For differences in unattached fractions, dose conversion coefficients were scaled by the ratio of the corresponding lung deposition fractions, which are proportional to the fraction of inhaled radon progeny transferred to the different organs. Note that lung doses were calculated by the IDEAL-DOSE model [19,39] for the specific exposure conditions in the three therapeutic facilities, i.e., no published dose conversion coefficients were required.

The basal and secretory cells of the bronchial and bronchiolar airways of the human lung receive by far the highest doses among all human organs, which demonstrates that lung cancer risk is the primary health effect related to radon-progeny inhalation. While IDEAL-DOSE predicts a lung dose of 5.01 mSv for the thermal gallery, corresponding calculations with the ICRP [44] lung model result in a lung dose of 6.64 mSv, consistent with earlier observations [20]. The doses for the liver are also representative of stomach, small intestine, colon, gonads, brain, bladder, and muscle. For comparison, organ doses calculated by Brudecki et al. [24], based on an updated biokinetic model and applying slightly different exposure conditions, are on average slightly higher by about 16% than the organ doses based on Kendall and Smith [25].

Dose calculations for the skin were based on the inhalation dose coefficient of Brudecki et al. [24] as the biokinetic model of Kendall and Smith [25] does not contain a skin compartment. Since no information has been published on the distribution of radon progeny in the skin, the same doses were assumed for both Langerhans and basal cells.

### 3.4. Radon-Progeny Attachment to Skin

Measurements of radon-progeny activities accumulated on skin surfaces in thermal and vapor baths, as well as of their depth distributions across the upper layers of the skin, have already been reported in a previous study [29,30]. These surface activities refer to radon activity concentrations of 950 kBq m^−3^ in thermal water and 90 kBq m^−3^ in chamber air at the time of their measurements. While an equilibrium factor of 0.11 was measured in the exposure chamber, no experimental information of the unattached fraction f_p_ is currently available. Since radon-progeny exposure conditions in the thermal gallery are similar to those in the vapor bath [45,48,49], the unattached fraction of 4% measured in the thermal gallery was also assumed for the vapor bath. Thus, radon-progeny surface activities in the thermal gallery were extrapolated from the surface activities measured in the vapor bath by scaling the corresponding radon activity concentrations, equilibrium factors, and exposure times.

An experimental comparison between the thermal gallery and vapor bath was conducted by Just [49], who measured a ratio of about 1.7 between the surface activities for radon activity concentrations in the vapor bath and the thermal gallery, normalized to the same radon activity concentration and exposure time. For comparison, this normalized ratio in the present study was 2.5. In a similar study, radon-progeny surface activities measured by Falkenbach et al. [44] with a Geiger–Mueller counter were higher by about a factor of 1.3 than our extrapolated values. This general agreement with available experimental data confirms the validity of the present extrapolation of surface activities from the vapor bath to the thermal bath.

For subsequent dose calculations, Langerhans cells (immune response) were assumed to be uniformly distributed throughout the epidermis from 15 to 45 μm, and basal cells (skin cancer) were located at a median depth of 50 μm [57]. Doses to Langerhans and basal cells resulting from the attachment of radon progeny to the skin in the thermal bath, vapor bath, and thermal gallery are compiled in Table 5, based on average radon-progeny surface activities for males and females.

Annual basal-cell doses for radon progeny deposited on the skin have also been published by Harley and Robbins [58] and Kendall and Smith [25] based on an environmental radon activity concentration of 200 Bq m^−3^. For a radon activity concentration in the thermal gallery of 50 kBq m^−3^ and a total exposure time of 10 h, basal-cell doses range from 2.8 mSv for stagnant air [58] to 7.1 mSv for moving air [25]. The lower value of 2.1 mSv in the present study may have been caused by a lower deposition velocity of radon progeny in humid air as compared to an outdoor environment.

Measurements by Falkenbach et al. [45] and Tempfer et al. [30] revealed significant differences among volunteers as well as among the measured parts of the human body (forehead, forearm, abdomen, and lower leg). Furthermore, the bodies of the patients were partly covered by a bathing suit or swimming trunks and patients were lying on their back as in the thermal gallery.

### 3.5. External Irradiation

In the case of the thermal gallery, gamma rays emitted primarily from the surrounding radium-containing rocks may have contributed to the resulting organ doses. For example, measured external dose rates of about 0.7 μGy h^−1^ lead to a skin dose of 7 μSv for 10 treatments, and about 4 μSv for all inner organs. Thus, organ doses produced by external gamma rays were negligible compared to alpha doses and thus may safely be neglected for dosimetric purposes.

## 4. Comparison of Exposure Pathways

The radon and radon-progeny exposures in thermal water comprise radon uptake through the skin and radon-progeny attachment on the skin, supplemented by radon and radon-progeny inhalation. The comparison of organ doses incurred in the thermal bath via the four exposure pathways is presented in Table 6. The apparent dominant modes of exposure are the attachment of radon progeny to the skin of the patients, producing the highest doses in the Langerhans and basal cells of the skin, and the uptake of radon from the thermal water through the skin in the lungs and red bone marrow (RBM). For comparison, inhalation of radon and radon progeny plays only a minor role as a result of the low ambient activity concentrations in the well-ventilated therapy rooms. Thus, thermal water is the responsible factor for the therapeutic effect via radon-progeny attachment and radon uptake through the skin.

The comparison of organ doses received in the vapor bath for the four exposure pathways illustrated in Table 7 reveals a similar result, again with the highest doses for the Langerhans and basal cells due to the attachment of radon progeny on the skin. Although the head of the patient remained outside of the exposure chamber, the opening for the head was not tight enough to prevent radon from escaping from the exposure chamber, thereby producing a significant dose to the lungs. As in the thermal bath, the attachment of radon progeny on the skin would have been primarily responsible for the therapeutic response.

The distribution of doses among the organs and tissues of the human body resulting from the different exposure pathways in the thermal gallery is displayed in Table 8. Due to high radon-progeny activity concentrations in the inhaled air and the longer exposure times, the lungs received the highest dose, amounting to about 5 mSv. Although the attachment of radon progeny on the skin led to doses comparable to that of the lungs, the partial protection of the body by swimwear and the supine position of the patients on the beds may have reduced the skin doses to about 60–70% of the calculated values. Although radon inhalation and radon transfer through the skin were based on the same radon activity concentration, the resulting organ doses differed by about an order of magnitude. The lower values for the radon uptake via the skin may be attributed to the slower and less effective transfer through the skin as compared to the fast uptake and immediate transfer to the blood via the lungs. Thus, radon-progeny deposition on the skin and, to a lesser extent, radon and radon-progeny inhalation, were the dominant contributors to a potential therapeutic response.

In conclusion, the primary exposure pathways in the three treatment facilities in the Gastein radon spas are (i) radon-progeny deposition on the skin and radon uptake through the skin in the thermal bath, (ii) attachment of radon progeny on the skin in the vapor bath, and, (iii) radon-progeny deposition on the skin as well as radon and radon-progeny inhalation in the thermal gallery.

## 5. Discussion and Conclusions

In the present study, doses were calculated for selected organs and tissues of the human body, comprising the lungs, red bone marrow, liver, kidneys, and the Langerhans and basal cells of the skin. While incurred doses in red bone marrow, liver, kidneys, and the Langerhans cells in the skin may be correlated to potential therapeutic effects, doses to the lungs and the basal cells of the skin indicate potential carcinogenic effects. Although not listed in the tables, doses to the kidneys are also representative of the doses to stomach, small intestine, colon, gonads, brain, bladder, and muscle. Complementary information for doses to other organs can be found in the original publications of Khursheed [11] and Sakoda et al. [12] for radon inhalation, of Sakoda et al. [13] for radon uptake through the skin, and of Kendall and Smith [25] and Brudecki et al. [24] for radon-progeny inhalation.

The highest doses among the different exposure pathways and treatment facilities were delivered to the lungs by the inhalation of radon progeny and to Langerhans and basal cells of the skin by the attachment of radon progeny to the skin surfaces, except for the thermal bath, where the transfer of radon through the skin approached the values for the radon-progeny attachment. Thus, from a therapeutic perspective, the most important exposure pathways were the radon progeny attached to the skin in all three treatment facilities, the transfer of radon through the skin in the thermal bath, and radon and radon-progeny inhalation in the thermal gallery. With respect to the thermal gallery, the significant contribution of radon-progeny attachment to the skin questions previous hypotheses that the observed therapeutic effects were solely attributed to the inhalation of radon and radon progeny.

Organ doses in this study represented average doses, based on typical average radiological and biokinetic parameters, and therefore ought to be considered only as best estimates. Indeed, significant temporal fluctuations of the radiological parameters, such as the radon activity concentrations in the vapor bath and thermal gallery, as well as individual variations of anatomical and physiological parameters, such as respiratory parameters or body dimensions, have been observed [15,30]. In particular, the experiments of the radon transfer through the skin and the radon-progeny attachment to the skin were performed with volunteers between the ages of 26 and 40 years, while the majority of patients receiving radon treatment are typically much older. Additional factors to be considered are gender differences with respect to the biokinetic parameters and organ volumes and masses in the biokinetic models [11,12,13,14,24,25,26] as well as experimentally observed differences in radon transfer rates and radon-progeny attachment rates [15,30]. As a result, doses for a given patient at a given time of exposure for the different exposure pathways may deviate from the organ doses presented in this study.

While dose calculations for the transfer of radon through the skin in thermal water and the vapor bath, and the deposition of radon progeny on the skin in water and air, were based on experimental studies with several volunteers [15,30], corresponding dose calculations for radon and radon-progeny inhalation were derived from published dosimetry models [11,12,24,25]. Since these dosimetry models provide information on annual dose conversion coefficients for the continuous inhalation of radon and radon progeny, the assumption of steady-state conditions, except for the lungs, may slightly overestimate the resulting doses. For comparison, dose calculations for radon uptake through skin and radon-progeny attachment to the skin do consider the temporal evolution of dose delivery. Due to the application of slightly different biokinetic models in the studies of Khursheed [11] and Sakoda et al. [12] for radon inhalation, and of Kendall and Smith [25] and Brudecki et al. [24] for radon-progeny inhalation, resulting dose conversion coefficients reveal slight differences between the various model predictions. However, these differences do not appreciably affect the present conclusions about the relative contributions of the different exposure pathways in the three therapeutic regimes.

In the present study, doses resulting from the inhalation of radon progeny were based on the dose conversion coefficients of Kendall and Smith [25] for red bone marrow, liver, and kidneys, and of Brudecki et al. [24] for the skin. For comparison, the recently published ICRP Report 147 [59] recommended a significantly higher dose conversion coefficient for the effective whole-body dose, which implies that organ-specific dose conversion coefficients ought to be correspondingly higher. However, the ICRP Report 137 [59] does not provide any information about dose conversion coefficients for individual organs and tissues. Since the effective dose for radon-progeny inhalation is dominated by the dose to the lungs, the dose conversion coefficients for the remaining organs cannot reasonably be scaled relative to the effective dose for all organs. Fortunately, the contributions of radon-progeny inhalation to the total organ doses in the three radon facilities are small compared to the other exposure pathways. Thus, although doses to red bone marrow, liver, kidneys, and skin are likely higher than predicted in this study, these higher organ doses will not appreciably affect the resulting dose estimates for the thermal bath, vapor bath, and thermal gallery.

The primary objective of the present comparison of exposure pathways of radon and radon progeny in radon therapy was their relevance for the expected therapeutic response. For example, the Langerhans cells in the skin are supposed to stimulate an immune reaction [30], while red bone marrow, liver, and kidneys were selected to represent the overall organ response in radon therapy. On the other hand, inhalation of radon progeny in the lungs may cause bronchial carcinomas [60], and irradiation of the basal cells of the skin may give rise to skin cancer [31]. While the primary focus in radon therapy is understandably on the efficacy of a given therapeutic application, each exposure regime to radon and radon progeny is also closely associated with potential harmful effects. However, even in the case of the thermal gallery, where organ doses are highest among the three therapeutic applications, the annual effective dose for this specific treatment once a year lies within the statistical fluctuations of natural radiation exposure, ranging from 1 to 10 mSv [61]. For comparison, the staff working in the different spa locations, particularly in the thermal gallery, are subject to radiation-protection regulations.

In conclusion, the goal of the present study was to provide dosimetric information for selected organs and tissues for the three radon therapeutic regimes offered in the Gastein radon spas. Provided that therapeutic effects in radon therapy can be primarily attributed to the action of ionizing radiation, this dosimetric database can serve as a basis for the assessment of anticipated beneficial effects associated with the different exposure pathways and treatment facilities. However, additional biological and environmental factors may contribute to the interpretation of observed therapeutic response, such as the high humidity and temperature in the vapor bath and thermal gallery, or the mineral content of the water in the thermal bath. In addition, the high altitude of Gastein, fresh mountain air, or prescribed accompanying physical activities, may also beneficially contribute to the overall response.

## Figures and Tables

**Figure 1 ijerph-18-10870-f001:**
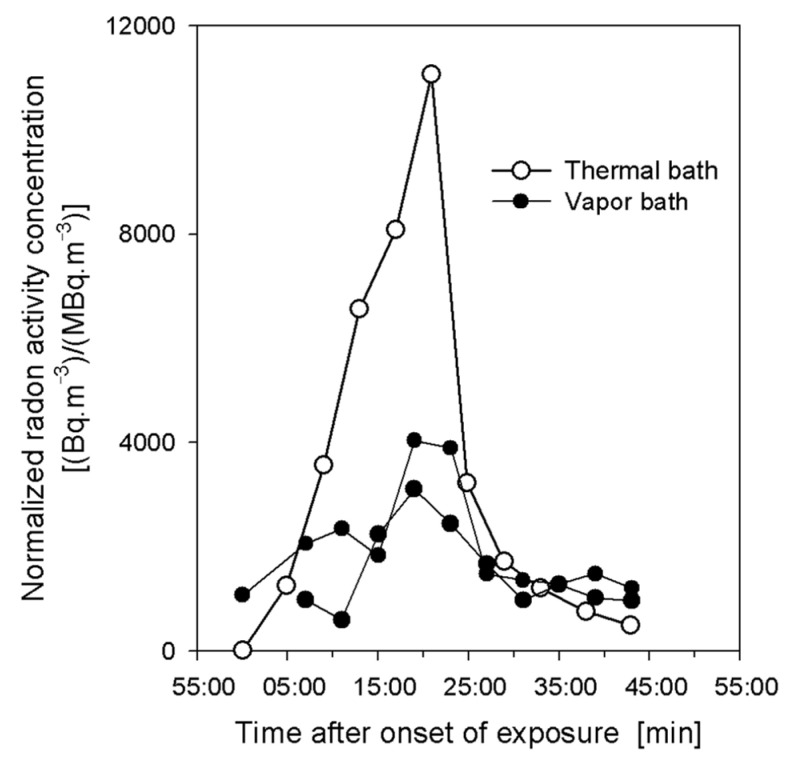
Exhalation curves for two measurements with a male volunteer in the vapor bath, normalized to the corresponding mean radon activity concentrations [Bq m^−3^/(MBq m^−3^)], for 20 min exposure and subsequent 20 min resting phases, and comparison with the corresponding exhalation curve obtained in the thermal bath with the same volunteer.

**Table 1 ijerph-18-10870-t001:** List of input parameters for the dose calculations in the thermal bath, vapor bath, and thermal gallery: Radon activity concentration (Rn), equilibrium factor (F), activity median diameters (AMD) for attached (a) and unattached (u) radon progeny, unattached fraction of the PAEC (f_p_), and Working Level (WL).

Input Parameters	Thermal Bath		Vapor Bath		Thermal Gallery
	Water	Room	Room	Chamber	
Rn [kBq m^−3^]	950	0.25	3.5	90.0	50.0
F	-	0.20	0.10	0.11	0.50
AMD (a) [nm]	-	290	230	230	230
AMD (u) [nm]	-	1.1	1.0	1.0	1.0
f_p_ [%]	-	25	8	4	4
WL	-	-	-	10.6	6.8

**Table 2 ijerph-18-10870-t002:** Selected organ-specific equivalent doses for lungs, red bone marrow (RBM), liver, and kidneys, resulting from the inhalation of radon in the thermal bath (250 Bq m^−3^), vapor bath (3.5 kBq m^−3^), and thermal gallery 50 kBq m^−3^), based on the dose conversion coefficients of Khursheed [11] and considering differences in breathing rates and exposure times. Doses to Langerhans cells (LC) and basal cells (BC) in the skin were calculated with the radon activity concentrations provided by Sakoda et al. [12].

Target Organ	Equivalent Dose [μSv]		
	Thermal Bath	Vapor Bath	Thermal Gallery
Lungs	0.33	4.67	166.73
RBM	0.19	2.59	91.92
Liver	0.03	0.35	12.42
Kidneys	0.02	0.18	6.53
Skin (LC)	0.02	0.30	3.61
Skin (BC)	0.02	0.30	3.61

**Table 3 ijerph-18-10870-t003:** Selected organ-specific equivalent doses for lungs, red bone marrow (RBM), liver, kidneys, and Langerhans cells (LC) and basal cells (BC) in the skin, resulting from the uptake of radon through the skin in the thermal bath, vapor bath, and thermal gallery [15].

Target Organ	Equivalent Dose [μSv]		
	Thermal Bath	Vapor Bath	Thermal Gallery
Lungs	333.90	10.45	17.45
RBM	185.25	5.80	9.69
Liver	25.65	0.80	1.34
Kidneys	14.25	0.45	0.75
Skin (LC)	25.80	0.81	1.35
Skin (BC)	25.80	0.81	1.35

**Table 4 ijerph-18-10870-t004:** Selected organ-specific equivalent doses for lungs, red bone marrow (RBM), liver, kidneys, and skin, resulting from the inhalation of radon progeny in the thermal bath, vapor bath, and thermal gallery, based on the dose conversion coefficients of Kendall and Smith [25], considering differences in breathing rates, exposure times, equilibrium factors, and unattached fractions. Lung doses were calculated with the IDEAL-DOSE model [19,39] and doses to Langerhans cells (LC) and basal cells (BC) in the skin with the inhalation dose coefficients of Brudecki et al. [24].

Target Organ	Equivalent Dose [μSv]		
	Thermal Bath	Vapor Bath	Thermal Gallery
Lungs	6.74	145.42	5006
RBM	0.01	0.152	4.67
Liver	0.01	0.25	7.78
Kidneys	0.13	2.73	83.97
Skin (LC)	0.01	0.07	2.65
Skin (BC)	0.01	0.07	2.65

**Table 5 ijerph-18-10870-t005:** Organ-specific equivalent doses for the Langerhans cells (LC) and basal cells (BC) in the epidermal layer of the skin, resulting from the attachment of radon progeny to the skin in the thermal bath, vapor bath, and thermal gallery.

Target Organ	Equivalent Dose [μSv]		
	Thermal Bath	Vapor Bath	Thermal Gallery
Skin (LC)	216	587	4447
Skin (BC)	118	299	2265

**Table 6 ijerph-18-10870-t006:** Comparison of organ doses received in the thermal bath for different exposure pathways.

Target Organ	Equivalent Dose [μSv]				
	Radon		Radon Progeny		Total
	Inhalation	Skin Uptake	Inhalation	Attachment	Dose
Lungs	0.33	333.90	6.74	-	340.97
RBM	0.19	185.25	0.01	-	185.45
Liver	0.03	25.65	0.01	-	25.69
Kidneys	0.02	14.25	0.13	-	14.40
Skin (LC)	0.02	25.80	0.01	216	241.83
Skin (BC)	0.02	25.80	0.01	118	143.83

**Table 7 ijerph-18-10870-t007:** Comparison of organ doses received in the vapor bath for different exposure pathways.

Target Organ	Equivalent Dose [μSv]				
	Radon		Radon Progeny		Total
	Inhalation	Skin Uptake	Inhalation	Attachment	Dose
Lungs	4.67	10.45	145.42	-	160.54
RBM	2.59	5.80	0.17	-	8.56
Liver	0.35	0.80	0.25	-	1.40
Kidneys	0.18	0.45	2.73	-	3.36
Skin (LC)	0.30	0.81	0.07	587	588.18
Skin (BC)	0.30	0.81	0.07	299	300.18

**Table 8 ijerph-18-10870-t008:** Comparison of organ doses received in the thermal gallery for different exposure pathways.

Target Organ	Equivalent Dose [μSv]				
	Radon		Radon Progeny		Total
	Inhalation	Skin Uptake	Inhalation	Attachment	Dose
Lungs	166.73	17.45	5006	-	5190.18
RBM	91.92	9.69	4.67	-	106.28
Liver	12.42	1.34	7.78	-	21.54
Kidneys	6.53	0.75	83.97	-	91.25
Skin (LC)	3.61	1.35	2.65	4447	4455
Skin (BC)	3.61	1.35	2.65	2265	2273

## Data Availability

Data sharing not applicable.

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
