# Peer review of "Dosimetric Comparison of Exposure Pathways to Human Organs and Tissues in Radon Therapy"

_ijerph, 2021, doi:10.3390/ijerph182010870_

Round 1

Reviewer 1 Report

  1. The authors point to publications describing the positive effects of radon therapy in spas. However, there are also separate opinions, and for the sake of balance it would be good to indicate these views as well.
  2. Chapter 3.3. According to the newest recommendation of the ICRP (Report 137), the dose conversion factors are significantly higher. Therefore it would be interesting to provide in this article the explicit values of the conversion factors you have used to assess the dose corresponding to the short-lived radon progeny.
  3. Lines 45 -46, 51. Maybe it is from the context. However, please specify what concentration is it, in what medium? Just for the sake of the readers. The radon concentration inside the exposure chamber was measured after taking the air sample to the Lucas cells via a dryer with silica gel ?
  4. Line 96. though -> through
  5. Lines 150-151. This sentence is somewhat misleading. At the very beginning I thought about the transfer in terms of the normalized values. It would be better: “ the radon transfer during the treatment”

Reviewer 2 Report

The authors compared radiation does through three radon therapies, i.e. (i) exposure to radon in a thermal bath, (ii) exposure to radon vapor, and (iii) exposure to radon in the thermal gallery. The results are unique and worth reporting, but the reviewer feels the manuscript does not have enough scientific descriptions. Detail descriptions about participants in this experiment are required, including ages, sex, ethical consents obtained.
Moreover, more numerical results must be mentioned in the abstract..

Round 2

Reviewer 2 Report

The authors revised properly, and the reviewer is now satisfied with their modification.